# Repeated-Dose Pharmacodynamics of Pimobendan in Healthy Cats

**DOI:** 10.3390/ani12080981

**Published:** 2022-04-11

**Authors:** Keisuke Sugimoto, Kazutoshi Sugita, Kensuke Orito, Yoko Fujii

**Affiliations:** 1Faculty of Veterinary Medicine, Okayama University of Science, Ehime 7948555, Japan; 2Faculty of Veterinary Medicine, Azabu University, Kanagawa 2525201, Japan; sugita@azabu-u.ac.jp (K.S.); oritok@azabu-u.ac.jp (K.O.); fujiiy@azabu-u.ac.jp (Y.F.)

**Keywords:** cardiac systolic function, feline, heart failure, hypertrophic cardiomyopathy

## Abstract

**Simple Summary:**

The dosing of pimobendan in cats is determined with reference to the effects of a single dose, although pimobendan is normally administered in repeated doses. In this study, the pharmacodynamics of repeated and multiple-dose pimobendan in healthy cats was investigated. Data were collected from five cats. Cardiac systolic function increased after repeated-dose pimobendan administration and correlated well with plasma concentrations of the drug. The results of this study suggest that a higher dose of pimobendan is effective.

**Abstract:**

The aims of this study were to investigate the effects of repeated and multiple-dose pimobendan on cardiac systolic function and the correlations between changes in cardiac systolic function and plasma concentrations of pimobendan and O-desmethylpimobendan (ODMP). Five clinically healthy cats were subjected to four different medication protocols for 14 days, with a washout period of at least 1 month between each protocol. The protocols were pimobendan 0.5 mg/kg q12h (high dosage [HD] group); pimobendan 0.25 mg/kg q12h (standard dosage [SD] group); pimobendan 0.125 mg/kg q12h (low dosage group); and Biofermin R, one tablet q12h (placebo group). Echocardiography and measurement of plasma concentrations of pimobendan and ODMP were performed prior to medication administration (baseline) and 20, 40, 60, 120, 240, 360, and 480 min after administration, and the correlation between the changes in cardiac systolic function and plasma concentration of pimobendan, ODMP, or the sum of both were examined. The cardiac systolic function increased in the HD and SD groups, and there were significant correlations between the lateral peak systolic myocardial velocity (S′) changes and plasma pimobendan, plasma ODMP, and the sum of both. Repeated doses of pimobendan in healthy cats increased cardiac systolic function, and there were significant correlations between cardiac function and plasma concentrations of pimobendan and ODMP. The results of this study highlight the effectiveness of a higher dose of pimobendan.

## 1. Introduction

Pimobendan is a calcium sensitizer and phosphodiesterase III inhibitor that acts both as a positive inotrope and a balanced vasodilator [1,2]. Following its administration, pimobendan is metabolized to an active metabolite, O-desmethylpimobendan (ODMP), which induces similar inhibitory effects, primarily through phosphodiesterase III inhibition [3,4,5]. Pimobendan is used in the treatment of dogs with myxomatous mitral valve disease (MMVD) and dilated cardiomyopathy (DCM) [6]. In dogs with cardiac remodeling or chronic heart failure (CHF) caused by MMVD or DCM and preclinical DCM, pimobendan is recommended at a standard dosage [7,8,9,10,11]. In dogs with end-stage MMVD, the effectiveness of an additional dose of pimobendan is suggested, although it is considered as off-label use [7].

The administration of pimobendan for cats has also been reported. Schober et al. [12] reported that pimobendan (0.30 mg/kg q12h) had no benefit on the 180-day outcome in cats with hypertrophic cardiomyopathy (HCM) and recent CHF. In contrast, several reports have indicated the effectiveness of pimobendan in cats. Reina-Doreste et al. [13] reported that pimobendan (0.075–0.5 mg/kg q12h) had a significant benefit in survival time for cats with HCM with CHF. Gordon et al. [14] reported that pimobendan (0.26 ± 0.08 mg/kg q12h) was well tolerated in cats with left ventricular systolic dysfunction. The dosage of pimobendan was referenced in two studies [15,16]. Both studies reported the effects of single-dose pimobendan on blood concentration and cardiac function, although pimobendan is a repeated-dose drug. Yata et al. [15] reported the treatment effects of a single dose and two doses of pimobendan (0.14 and 0.28 mg/kg) on cardiac function. However, Hanzlicek et al. [16] reported that the maximum plasma concentration of pimobendan on the third day was lower than that measured on the first day of pimobendan administration (0.28 ± 0.04 mg/kg q12h). Furthermore, in human patients with CHF the plasma concentrations of pimobendan and ODMP, as well as cardiac function, were decreased when the drugs were administered in repeated doses compared with a single dose [17]. The effects of repeated-dose pimobendan on cardiac function in cats have not been previously reported. This study aims to investigate the effects of repeated and multiple-dose pimobendan on cardiac systolic function and the possible correlations between changes in this parameter and the plasma concentrations of pimobendan and ODMP.

## 2. Materials and Methods

### 2.1. Animals

Five clinically healthy cats (two males, three females) were included in this study. The cats were all mixed breed, aged between four and seven years and weighting 2.9–4.8 kg. A complete physical examination, complete blood count, blood chemistry, echocardiography, thoracic radiography, and blood pressure measurements were performed in all cats to exclude systemic diseases. None of the cats had abnormal findings. The cats were kept individually in cages, fed with commercially available cat food, and had free access to water.

### 2.2. Experimental Protocol

The experimental protocol was shown in Figure 1. Five cats were administered pimobendan for 14 days, and echocardiography and measurement of plasma concentration were performed on day 15.

All five cats were subjected to each of the following four medication protocols for 14 days: pimobendan (DS Pimo heart tablet, DS Pharma Animal Health, Osaka, Japan) 0.5 mg/kg q12h (high dosage [HD] group), pimobendan 0.25 mg/kg q12h (standard dosage [SD] group), pimobendan 0.125 mg/kg q12h (low dosage [LD] group), and Biofermin R (probiotic) (Biofermin Pharmaceutical Inc., Osaka, Japan), one tablet q12h (placebo group).

In each protocol, all medications were administered orally at 8:00 AM and 8:00 PM, and 3 mL of water were administered orally following the pill to ensure ingestion. The cats were fed 30 min after the administration of the medication. Echocardiography and measurements of plasma concentrations of pimobendan and ODMP were performed on the 14th day of each protocol. There was a washout period of at least 1 month between protocols. All cats underwent each protocol in the following order: HD, SD, placebo, and LD.

### 2.3. Cardiac Systolic Function

Cardiac systolic function was assessed using M-mode and color tissue Doppler imaging (TDI). Echocardiography was performed prior to the medication administration (baseline) and 20, 40, 60, 120, 240, 360, and 480 min after administration. All echocardiographic images were acquired using an ultrasound unit equipped with a 7 MHz transducer (Vivid 7 dimension, GE Medical System, Tokyo, Japan). Echocardiographic examination was performed by K. Sugimoto. Off-line image analysis was performed using commercial software (EchoPAC PC, GE Medical System, Tokyo, Japan), and the mean values of variables in five consecutive cardiac cycles were used for statistical analysis.

The cats were gently restrained in lateral recumbency without sedation. Left ventricular (LV) end-diastolic diameter and LV end-systolic diameter were measured at the right parasternal short-axis chordae tendineae, and the LV fractional shortening (FS) was calculated [18]. The heart rate (HR) was calculated from M-mode images. All color TDI examinations were performed as described previously [19]. The lateral aspect of the mitral annulus was sampled using the left four-chamber apical view. A 2 × 2 mm sample volume without angle correction was used. The lateral peak systolic myocardial velocity (S′) was also measured.

### 2.4. Plasma Concentration of Pimobendan and O-Desmethylpimobendan

Blood samples (1 mL each) were collected from the medial saphenous vein at the time at which the changes in cardiac systolic function reached a maximum, and were placed into tubes with the anticoagulant EDTA. All samples were kept for 10 min and centrifuged to obtain plasma (3000 rpm, 10 min), and stored at or below −80 °C for 4 to 6 months before analysis. Plasma concentrations of pimobendan and ODMP were determined using liquid chromatography-tandem mass spectrometry. The recoveries of pimobendan and ODMP were 106 ± 4.2% and 113 ± 0.3%, respectively.

### 2.5. Pharmacodynamics Analysis

To evaluate the pharmacodynamics of pimobendan, cardiac systolic function at baseline was compared among the groups to assess the effect of repeated administration, and cardiac systolic function was also compared among the groups. Furthermore, the correlation between changes in cardiac systolic function and plasma concentration of pimobendan, ODMP, or the sum of pimobendan and ODMP were analyzed. The changes in cardiac systolic function were calculated as the differences between the baseline value and the value at the time at which cardiac systolic function reached a maximum. Individual correlations were also confirmed.

### 2.6. Statistical Analyses

All measurements are expressed as mean ± standard deviation. Statistical analyses were performed using commercial computer software (EXCEL toukei version 7.0, Esumi Co., Ltd., Tokyo, Japan).

All echocardiographic data were visually inspected and tested for normality using the Kolmogorov–Smirnov test. When a significant difference was detected, multiple comparisons were performed using Bonferroni’s multiple comparison test. Cardiac systolic function was also compared using two-way analysis of variance (ANOVA) with dosage and time as independent factors. When a significant difference was detected, multiple comparisons were performed using Bonferroni’s multiple comparisons at each time point. Cardiac systolic function at baseline was compared among the groups using one-way repeated measures ANOVA. The correlation between the changes in cardiac systolic function and plasma concentration of pimobendan, ODMP, or the sum of pimobendan and ODMP were examined using Pearson’s correlation coefficient. Intra-observer variability for cardiac systolic function was assessed by calculating the coefficient of variation (CV) using the following formula: CV = (SD/arithmetic mean of measurements) × 100 [20]. CV was considered clinically acceptable if it was below 10% [19]. Differences were considered statistically significant at *p* < 0.05.

## 3. Results

The results of cardiac systolic function are shown in Figure 2. There were no significant differences in FS, HR, and S′ at baseline among the groups. There were no significant differences in FS and HR at each time point, although visual inspection of the FS curves for the HD and SD groups appears to show an increase within the first few hours of treatment. There was a significant main effect of dosage (F(3128) = 2.34, *p* = 0.028) and time (F(7128) = 20.11, *p* < 0.01), and there was no significant interaction effect between dosage and time (F(21,128) = 1.00, *p* = 0.46). S′ at 60 and 120 min was significantly increased in the HD and SD groups compared with the LD and placebo groups, but there were no significant differences between the HD and SD groups or between the SD and placebo groups. The time at which the changes in cardiac systolic function reached a maximum in S′ was 60 or 120 min (three cats in the HD, four cats in the SD, and two cats in the LD group).

The results of the correlation between changes in S′ and plasma concentration of pimobendan, ODMP, and the sum of pimobendan and ODMP at 60 and 120 min are shown in Figure 3, Figure 4 and Figure 5. There was a significant positive correlation between them, although the plasma concentration was highly similar to that in the other cats. The correlation between the changes in S′ and plasma concentration at 60 or 120 min in individual cats is shown in Figure 6. In most cats, the changes in S’ were correlated with the plasma concentration of pimobendan and ODMP, although the plasma concentration showed individual differences, particularly in the HD group.

The intra-observer CV was clinically insignificant in all measurements (range: 4.1–8.5%).

There were no adverse effects observed in the present study.

## 4. Discussion

In this study, the maximum effects of cardiac systolic function were observed at 60 or 120 min after administration, and there were significant correlations between changes in S’ and plasma pimobendan, ODMP, and the sum of pimobendan and ODMP. In cats, both pimobendan and ODMP might be correlated with cardiac systolic function.

The plasma concentration had a wide range, particularly in the HD group. Yata et al. [15] reported that maximum plasma concentrations of pimobendan and ODMP were 24.7–34.7 and 8.1–14.2 and 41.7–70.9 and 11.3–21.8 in dosage corresponding to LD and SD in this study, respectively. Our results are in agreement with those reports. In our study, plasma concentration did not increase adequately in two cats (cats three and five). Furthermore, they reported that in one cat, the plasma concentrations were significantly lower than the concentrations observed in other cats, although the cat had no abnormalities. Hanzlicek et al. [16] reported that in one cat, the plasma concentrations did not increase because of vomiting immediately after administration. In this study, no cats had abnormalities during the study. Pimobendan conversion to ODMP involves specific cytochrome P-450 isoenzymes in humans [21]. In addition, both pimobendan and ODMP are ultimately glucuronidated and excreted [22]. Differences in the activity of cytochrome P-450 isoforms and their ability to glucuronidate compounds have been identified in cats [23,24]. A population-based pharmacokinetic study in a larger number of individuals may aid in the further evaluation of this anomaly.

There were no significant differences in S′ between the HD and SD groups. Plasma concentration in some cats in the HD group did not increase adequately, although the plasma concentration did not increase as described above. If cardiac systolic function does not increase after administration of pimobendan, a higher dose of pimobendan may be effective in increasing cardiac systolic function.

In this study, S′ was significantly increased in the HD and SD groups, but not in the LD group. Previously, Yata et al. reported that pimobendan at equivalent dosage in the LD group in this study increased FS [9]. An increase in HR was correlated with an increase in FS. Myocardial contractility increases with an increase in HR, and a decrease in HR has a negative staircase effect [12]. In this study, there were no significant differences in the HR. Cardiac systolic function may not increase with repeated low-dose pimobendan.

There were no significant differences in FS and S′ at the baseline. Cardiac systolic function in repeated-dose pimobendan at baseline was significantly higher than that in single-dose pimobendan at baseline in human patients with DCM and CHF [25]. Healthy cats were used in the present study. Cardiac systolic function has been reported to be decreased in cats with cardiomyopathy or CHF [14,26,27,28]. Further studies are required to determine the effect of cardiac function in cats with cardiomyopathy or CHF.

S’ was remarkably increased in a single cat from the SD group at 120 min. This might be associated with an increase in HR. HR in this cat was sinus rhythm and 230 beats per minute at 120 min, but 150–190 beats per minute at other times. The experimental procedures were the same during the study; thus, the reason for the increased HR at this time point was unclear.

The present study has some limitations. First, the number of cats available was limited. Second, cats with normal cardiac function were used. The use of cats affected by heart disease with decreased cardiac function may offer different results from our study. Third, the protocols were not randomized, and the assessors were not blinded to the treatment protocols. Fourth, this study was focused on measuring cardiac systolic function, which reflects the primary effect of pimobendan. However, ventricular diastolic dysfunction and atrial function play a large role in feline heart disease due to the propensity of this species to develop HCM [29,30]. Therefore, the possible effects of pimobendan on ventricular diastolic and atrial functions require further evaluation.

## 5. Conclusions

Repeated doses of pimobendan in healthy cats increased cardiac systolic function, and there were significant correlations between cardiac function and plasma concentrations of pimobendan and ODMP. Furthermore, the results of this study suggest the effectiveness of a higher dose of pimobendan. These findings can be used to guide further investigations into the safety and efficacy of pimobendan treatment in feline patients.

## Figures and Tables

**Figure 1 animals-12-00981-f001:**
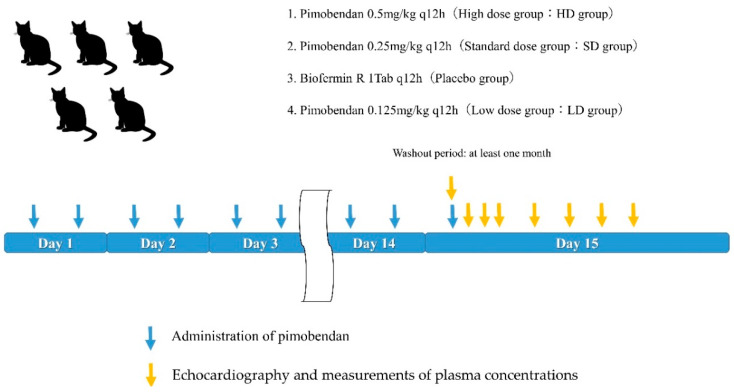
Scheme of the experimental timeline.

**Figure 2 animals-12-00981-f002:**
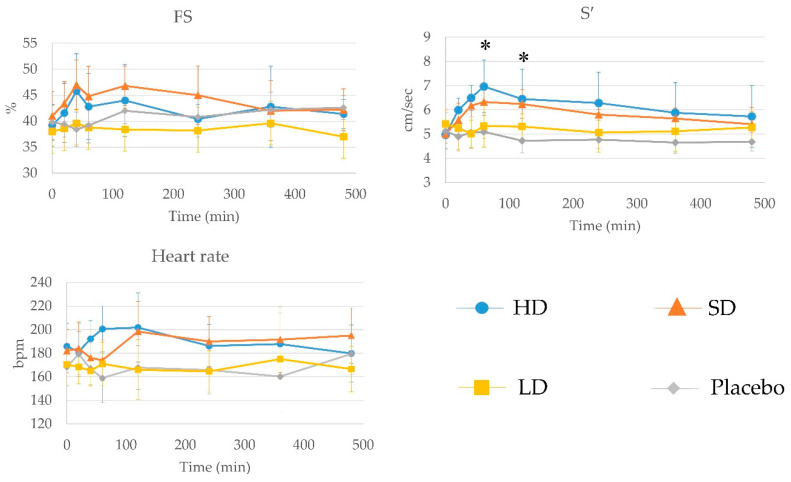
The results of cardiac systolic function. Values for lateral peak systolic myocardial velocity (S′) at 60 and 120 min in the HD and SD groups were significantly increased compared with the LD and placebo groups. *: Significant differences in the HD and SD groups compared with the LD and placebo groups (*p* < 0.05).

**Figure 3 animals-12-00981-f003:**
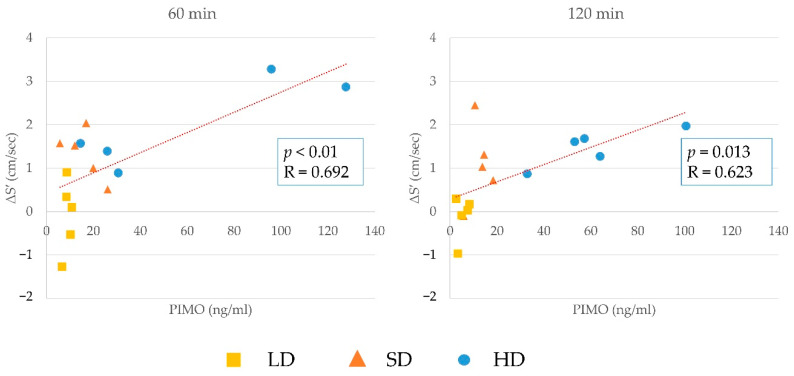
The results of the correlation between changes in lateral peak systolic myocardial velocity (S′) and plasma concentration of pimobendan at 60 and 120 min. There was a significant positive correlation between these values at both time points.

**Figure 4 animals-12-00981-f004:**
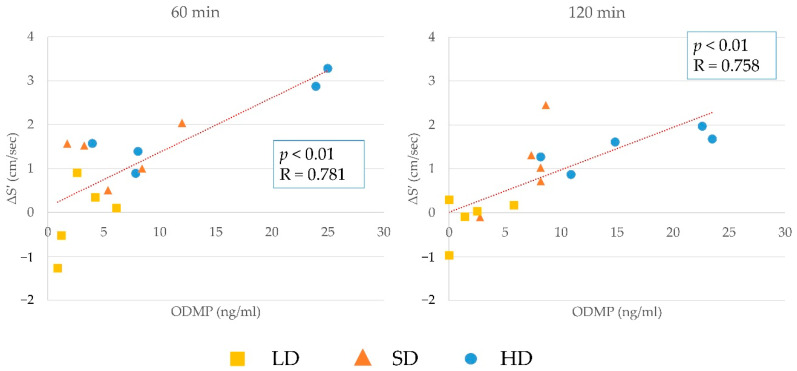
The results of the correlation between changes in lateral peak systolic myocardial velocity (S′) and plasma concentration of ODMP at 60 and 120 min. There was a significant positive correlation between these values at both time points.

**Figure 5 animals-12-00981-f005:**
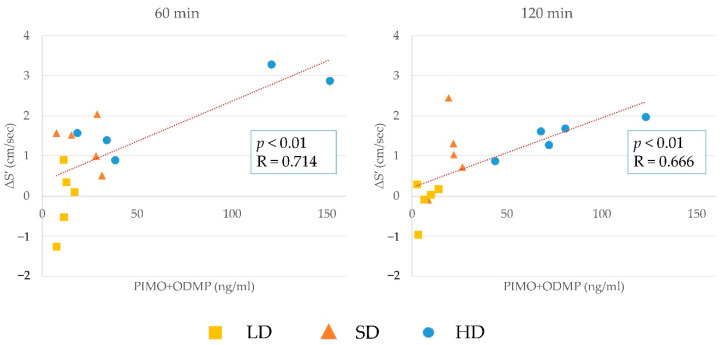
The results of the correlation between changes in lateral peak systolic myocardial velocity (S′) and plasma concentration of the sum of pimobendan and ODMP at 60 and 120 min. There was a significant positive correlation between these values at both time points.

**Figure 6 animals-12-00981-f006:**
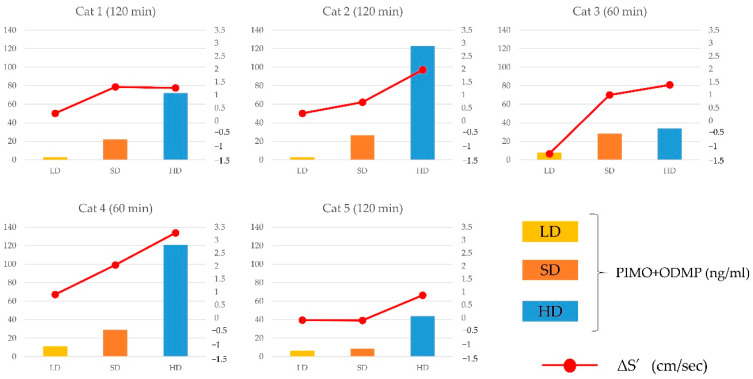
The correlation between the changes in lateral peak systolic myocardial velocity (S′) and plasma concentration of the sum of pimobendan and ODMP at 60 or 120 min in individual cats. In most of the cats, changes in S’ are correlated with the plasma concentration of pimobendan and ODMP.

## Data Availability

The data presented in this study are available on request from the corresponding author.

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
