# Peer review of "Repeated-Dose Pharmacodynamics of Pimobendan in Healthy Cats"

_animals, 2022, doi:10.3390/ani12080981_

Round 1

Reviewer 1 Report

This submission sought to review pimobendan (3 doses) in healthy felines without cardiac abnormalities to access PD outcomes on systolic function relative to dose. The paper is well written, particularly the discussion section where all elements of the study are addressed. Systolic function is altered in a dose dependent fashion.   

Line 26-27 – Echo + Blood were collected for baseline (time zero) “and 20, 40, 60, 120, 240, 360, and 480 minutes after administration” Were measurements taken on day 14 or 15 after receiving drug course?  This might be made clear with the addition of a simple scheme showing the experimental timeline for an example round of treatment (inserted near Line 77)

Line 71-73 – What about urinalysis?

Line 81 – Please include what Biofermin R (probiotic) is commonly used for as most readers will not know.

Line 110 – What anticoagulant? EDTA, Aprotinin, Sodium citrate, etc.

Line 165 – “There were no adverse effects observed…” – Vomiting in response to being medicated is adverse – Line 200 – “vomiting immediately after administration.” Please mention here as well as GI disturbance are a known side effect from Pimo.  

Figure 1 - reads blurry – Could be formatting issue with PDF copy or authors might want to consider a higher resolution figure – with focus on improving x/y axis labels for readability.

Line 169 – Figure legend should include statistical details to what * symbol corresponds to in terms of p-value.

Author Response

Line 26-27 – Echo + Blood were collected for baseline (time zero) “and 20, 40, 60, 120, 240, 360, and 480 minutes after administration” Were measurements taken on day 14 or 15 after receiving drug course?  This might be made clear with the addition of a simple scheme showing the experimental timeline for an example round of treatment (inserted near Line 77)

Response: Thank you for your recommendation. We added the scheme for the experimental timeline (Figure 1).

Line 71-73 – What about urinalysis?

Response: We did not perform urinalyses, because there were no abnormalities in blood chemistry or blood pressure, and no cats presented clinical indications.

Line 81 – Please include what Biofermin R (probiotic) is commonly used for as most readers will not know.

Response: Thank you for your suggestion. The related content has been added, accordingly.

Line 110 – What anticoagulant? EDTA, Aprotinin, Sodium citrate, etc.

Response: We used EDTA and have added the appropriate term, accordingly.

Line 165 – “There were no adverse effects observed…” – Vomiting in response to being medicated is adverse – Line 200 – “vomiting immediately after administration.” Please mention here as well as GI disturbance are a known side effect from Pimo. 

Response: The side effect in Line 200 was an observation in a previous study, not observed in our study. The potentially confusing statement has been removed.

Figure 1 - reads blurry – Could be formatting issue with PDF copy or authors might want to consider a higher resolution figure – with focus on improving x/y axis labels for readability.

Response: Thank you for your suggestion. We revised Figure 2 (previously Figure 1) accordingly.

Line 169 – Figure legend should include statistical details to what * symbol corresponds to in terms of p-value.

Response: Thank you for your recommendation. We added the corresponding symbols, accordingly.

Reviewer 2 Report

The use of pimobendan in cats is controversial. This is due to the fact that cats mainly suffer from diastolic dysfunction rather than systolic diseases for which pimobendan is dedicated. In addition, studies have shown that the response to pimobendan in this species is individual and therefore the effect of its use cannot be fully predicted.

In my opinion, the research carried out was done on too few animals to be able to draw conclusions from it. Even in such a small group, large individual differences can be seen, e.g. between cat 4 and cat 3.

In addition, it is known that cats are very sensitive to stress and all manipulations cause an increase in HR, even translation from one side to the other during echocardiographic examination. Sometimes a change in position and an increase in HR make it impossible to assess the diastolic function because we can see fusion EA. That is why, in my opinion, such frequent tests (echo and blood sampling) also had an impact on HR in this case, and this, in turn, on contractile function. In addition, in my opinion, it is impossible to translate the results of this study into cats with systolic dysfunction in which we would like to use pimobendan.

Author Response

Response: Thank you for your suggestion.

The number of cats used in this study was small, which is mentioned in the revised manuscript as a major limitation. Particularly in the HD group, the results had large differences. Further study is warranted to investigate the effects of pimobendan on cardiac function.

In this study, HR was increased in some cats. However, there were no significant differences in HR in any group, and there were no significant correlations between S’ and increased HR. Hence, we were concerned that S’ was increased by the administration of pimobendan, although it may have been affected a bit by increased HR. 

Reviewer 3 Report

The manuscript presents a simple but highly useful clinical investigation

Author Response

Response: Thank you for your positive comments concerning our manuscript.

Round 2

Reviewer 2 Report

All comments included in the first review were taken into account. The results are clearly presented and thoroughly discussed.

Author Response

Thank you for your positive comments concerning our manuscript.